# Nursing Student Satisfaction with the Teaching Methodology Followed during the COVID-19 Pandemic

**DOI:** 10.3390/healthcare10040597

**Published:** 2022-03-22

**Authors:** Marta Carolina Ruiz-Grao, Sandra Cebada-Sánchez, Carmen Ortega-Martínez, Antonia Alfaro-Espín, Eduardo Candel-Parra, Francisco García-Alcaraz, Milagros Molina-Alarcón, Victoria Delicado-Useros

**Affiliations:** 1Nursing Faculty, University of Castilla-La Mancha, 02071 Albacete, Spain; marta.ruiz@uclm.es (M.C.R.-G.); carmen.ortega@uclm.es (C.O.-M.); antonia.alfaro@uclm.es (A.A.-E.); eduardo.candel@uclm.es (E.C.-P.); francisco.galcaraz@uclm.es (F.G.-A.); milagros.molina@uclm.es (M.M.-A.); 2Health and Social Research Center, University of Castilla-La Mancha,16071 Cuenca, Spain; 3Human Neuroanatomy Laboratory, University of Castilla-La Mancha, 02071 Albacete, Spain; 4Nursing & Society Research Group, University of Castilla-La Mancha, 02071 Albacete, Spain; 5Instituto de Investigación en Discapacidades Neurológicas (IDINE), University of Castilla-La Mancha, 02071 Albacete, Spain

**Keywords:** nursing education research, student satisfaction, higher education, education online, nursing students

## Abstract

Background: Halfway through the 2019–2020 academic year, the entire university system was affected by an exceptional situation caused by the COVID-19 pandemic. Online learning was globally implemented for all degrees to finish the course and to meet academic objectives. This unforeseen change in teaching and subsequent evaluations meant teachers and students had to invest significant effort. Student satisfaction is used to measure the evaluation of teaching/learning processes in higher education. Our objective was to know and compare the satisfaction of nursing students taught at a Spanish public university after making changes to the teaching methodology. Methods: A descriptive observational study that measures student satisfaction. Study population: 240 students registered in academic years 2019–2020 and 2020–2021 answered the survey. The survey contained 30 items answered on a Likert-type scale. The main variables: the learning methodology (online or blended) was the independent variable; student satisfaction was the dependent variable. Descriptive and bivariate analyses were performed. Results: A response rate between 37.4% and 41.2%. Overall satisfaction was 2.75 points (SD 0.56) and 2.94 points (SD 0.49) with online learning and bimodal learning, respectively (maximum score 4 points) (*p* < 0.004). Conclusions: Student satisfaction was moderate–high for both learning methodologies. Students found that the b-learning methodology was the most valued.

## 1. Introduction

University teaching has changed in recent decades as a result of the emergence and generalization of information and communication technologies (ICT). Digital platforms were set up almost two decades ago to favor virtual learning environments to not only support teaching with a marked face-to-face content, but to also favor courses, teaching and online training programs (online learning or e-learning) or bimodal or “mixed” training (“blended learning” or b-learning) [1,2,3].

In the evaluation and accreditation systems of university education, students’ point of view is considered to be of much interest. In particular, students’ perception of teaching methodologies is a systematically evaluated aspect in teaching–learning processes in higher education [1,4,5,6]. As students are the main users of university services and education addresses them, they are the best people to evaluate their university. Although students’ vision is partial [7], their views and perceptions serve as indicators of improvement, and also for developing academic programs and curricula. Similarly, the importance of their perceptions of the methodological tools and resources made available to them to train with (virtual platforms, document-access services, teaching equipment, access to practical training, etc.) has been confirmed [6].

Research suggests that student satisfaction is a complex concept and can be defined as a short-term attitude that results from the evaluation of their experience with the received education service [8]. Student satisfaction comprises several dimensions that can be grouped into facilities, teaching staff, teaching methods, environment, enrolment and support services. [8,9]. Some authors describe student satisfaction as a subjective evaluation of various outcomes and experiences associated with education [10]. Student satisfaction has also been identified as a significant indicator of students enjoying their studies [11] and an evaluative component in learning effectiveness assessments [12]. Many student satisfaction studies have recently been carried out in different countries to investigate the success of online learning in higher education [3,13,14,15,16,17,18,19].

Halfway through the academic year 2019–2020 (just after the second 4-monthly period began), the whole Spanish university system was hit by an exceptional situation: the COVID-19 pandemic [20]. The state of alarm and imposed confinement brusquely interrupted the possibility of being physically able to perform activities at Spanish universities. This meant shifting to exclusive online learning, which implied drastically and abruptly transforming university teaching. In this context, the objectives of the Ministry and the Conference of Rectors of Spanish Universities ensured that the academic year would be completed, and generally guaranteed quality for online learning and when evaluating different subjects [21].

For the academic year 2020–2021, work was conducted within a reference framework, which started by acknowledging the maximum quality of face-to-face teaching. It ended with a mixed system known as “adapted face-to-face learning”, which aimed to maintain online activities (meetings, tutoring and certain teaching activities), plus face-to-face theoretical and practical learning, by following the recommendations of the Spanish Ministries of Health and Universities [21].

Throughout 2020, a wide range of strategies was set up in nursing training centers to ensure that the training of these professionals followed international quality standards [15]. Many recent publications have evaluated methodologies, the impact of COVID-19 on the university and its influence on health professionals’ training [13,14,15,22,23,24].

It is well-known that generalizing ICT use is one of the objectives of the teaching innovation projects that have been included in the different national calls organized by Faculties of Nursing in recent years. The singularity of nursing studies, like others in the biohealth area, implies opting for classroom teaching as an overall teaching strategy by bearing in mind not only its clinical practical component, but also the importance of integrating theory and practice into its training. Nonetheless, a recently systematic review evidenced that online learning is no less effective than traditional learning for nursing students to acquire clinical skills [25]. Students positively evaluate the so-called blended learning (b-learning) methodologies [3], which constitute a feasible alternative to face situations in which exclusive classroom teaching is impossible given compulsory physical distancing to prevent COVID-19 from spreading, and due to other changes in using available equipment.

A complex scenario remains because the pandemic is persistent and unforeseeable. At universities, the perspective is to continue with not only blended systems, whose intention is to adapt and limit classroom training marked by the pandemic’s evolution and safety measures, but also with blended training systems (already applied during the academic year 2020–2021) by gradually including b-learning. Student satisfaction has become one of the most widely accepted quality dimensions in Higher Education evaluations [7]. For this reason, our main study objective was to identify and compare nursing degree student satisfaction with adopted e-learning and b-learning methodologies affected by the COVID-19 pandemic.

## 2. Materials and Methods

### 2.1. Design

This observational descriptive study compared nursing student satisfaction with the online and bimodal methodologies employed by a Spanish public university during two academic years: 2019–2020 and 2020–2021.

### 2.2. Population

Our study population was formed by students from the Faculty of Nursing of a public university during academic years 2019–2020 and 2020–2021 when the teaching methodology was affected by COVID-19 pandemic restrictions. The students affected by the transition from face-to-face to online teaching in the second semester of academic year 2019–2020 (*n* = 372) were included (from March 2020 to June 2020). The students in the first semester of 2020–2021, who worked with a mixed teaching modality (blended) (*n* = 245), were also included (from September 2020 to January 2021). In both groups, the studied subjects were theoretical and compulsory according to our study plan.

### 2.3. Sources of Information

Data collection was performed by sending a survey to evaluate student satisfaction at two time points: June 2020, with online university training (e-learning); January 2021, with b-learning by means of Moodle and a questionnaire devised by Google Forms. Surveys were freely and voluntarily completed.

The survey was completed by 139 students during the academic year 2019–2020 and 101 during 2020–2021, which gave a study population of 240 students. We are not certain that the decision to respond was conditioned by any characteristic associated with the variable under study (satisfaction). If the hypothesis of independence were true, the studied subset would be the equivalent to having been selected as a simple random sample. Data were processed according to the random sampling assumption to provide confidence intervals (CI) for measurements and to quantify statistical significance.

### 2.4. Variables under Study

The main studied variables were the learning methodologies, e.g., online (e-learning) or blended (b-learning), as the independent variable. The online methodology consisted of an exclusively telematic methodology that replaced classes, workshops, exams and face-to-face tutorials with telematic activities, and the Teams and Moodle platforms were mostly used. B-learning or the mixed methodology was based on virtual and face-to-face classes and student activities on platforms, and theoretical-practical workshops were carried out in small groups, mainly as face-to-face.

Student satisfaction was the dependent variable measured by a survey. It included 30 items about student satisfaction with university teaching. Items were grouped into five dimensions: subjects’ structure (3 items); teacher-related aspects (teaching performance) (9 items); aspects related to the contents of different subjects overall (10 items); communication-related aspects (2 items); aspects related to the virtual learning environment (6 items). The covariables were socio-demographic (age, gender, course) and those related to the availability of the technologies employed in e-learning. The original instrument called Questionnaire Satisfaction of University Students Towards Online Training (CUSAUF) was validated [26]. It had already been used [6,27,28] to evaluate student satisfaction with university e-learning and b-learning. It includes items answered on a Likert-type scale according to students’ level of agreement: “Very Low” (1 point), “Low” (2 points), “High” (3 points) and “Very High” (4 points). The survey was adapted to the purpose of this study by adding three items (30 items in all). The questionnaire dimensions were left as the original ones.

### 2.5. Statistical Analysis

A descriptive statistical analysis (absolute frequencies and percentages; central tendency and dispersion measures according to the studied variables) and a bivariate analysis (to compare the mean values between groups/courses by X2 test and ANOVA) were carried out. A comparison was made of student satisfaction with the two methodologies to be compared by accurate statistical testing (Student—Fisher’s t and ANOVA, or others) in accordance with available data normality. To assess the normality of our variables, we applied the normality test based on Kolmogorov-Smirnov and Shapiro–Wilk. About the existence of possible outliers, a decision was made not to eliminate them if there were only a few and did not affect the results. The instrument’s reliability was measured by Cronbach’s alpha. The adapted instrument’s reliability gave a Cronbach’s alpha value of 0.955 (Cronbach’s alpha value of 0.954 for online learning and a Cronbach’s alpha value of 0.956 for bimodal learning). Level of significance was set at *p* < 0.05. Data were analyzed by version 25 of the SPSS statistical package.

### 2.6. Ethical Aspects

Students freely and voluntarily participated after being informed about the survey. Data remained anonymous. Recommendations about personal data processing followed current Spanish legislation. The project obtained a favorable report from the Ethics Committee for Clinical Research from the Health Area. All the principles and norms about research matters were respected in the field to which this project was applied.

## 3. Results

During the first academic year (March–June 2020), 139 valid surveys were collected (37.4% response rate), and 101 (41.2% response rate) in February 2021 (September 2020–February 2021). Table 1 shows the participants’ socio-demographic data and characteristics.

The bivariate analyses revealed a significant difference in the relation between the variables academic course and received learning type (*p* = 0.001) (Table 1).

The mean student satisfaction with bimodal learning was higher than for online learning (*p* < 0.004) (Table 2). However, no difference was observed in mean student satisfaction according to the learning methodology when the categorized variable was used for the high, low and moderate satisfaction levels (*p* = 0.071). More students indicated feeling moderate satisfaction with online learning (55.4%) and bimodal learning (65.3%) (Table 2).

Regarding the dimensions on the satisfaction scale, differences were observed between teaching methodologies for the dimension satisfaction with subjects’ structure (*p* < 0.001), teacher-related aspects (*p* < 0.007) and aspects related to the contents of the different subjects on the whole (*p* < 0.001) (Table 2).

The relation between satisfaction with the categorized online learning methodology and students’ gender showed marked differences (*p* = 0.003) (Table 3). An association appeared between students’ academic course and the categorized student satisfaction level for the bimodal learning/b-learning methodology (*p* = 0.002) (Table 3).

Finally, Table 4 shows large differences in the overall satisfaction scores given to the different courses for bimodal learning (*p* = 0.001), but not for online learning. For the satisfaction level per dimension, a difference was observed only for the online methodology in the different courses, with scores for aspects related to the virtual learning environment dimension. Differences were found for the b-learning methodology in the scores given to the different courses for all the evaluated dimensions, except for communication-related aspects (*p* = 0.093) (Table 4). In the survey items, some marked differences were found according to the followed methodology (Appendix A).

## 4. Discussion

The main objective of the present study was to know university nursing degree student satisfaction levels with the learning methodologies followed during academic years 2019–2020 and 2020–2021, which were affected by COVID-19 pandemic restrictions. These students indicated a moderate satisfaction level. In general, healthcare professionals’ training has had to face COVID-19 pandemic restrictions, plus the challenge of continuing or interrupting clinical training in health centers, bearing in mind the urgent need for graduating professionals who were very much in demand and required for the health crisis [29,30,31,32,33,34].

This study observed that of the followed methodologies, bimodal learning was better valued with a higher satisfaction level. When we wished to compare our results to those of other studies, we found that very few studies include and compare satisfaction with university online and mixed methodologies, and there are very few publications about student satisfaction in Europe to the best of our knowledge. Some consulted works center on evaluating satisfaction with online learning [16,19,20,21,30,31,32], some evaluate b-learning [15,22], and others assess satisfaction by comparing current methodologies such as online, bimodal and face-to-face [3,13].

Evaluating overall satisfaction with online learning encountered marked differences in the reviewed research works. Some studies report that most students were unsatisfied as follows: 42% of university medical and nursing students in a study conducted in India [13]; 48.6% of the students studying different disciplines in a work from the Lebanon [14]; in Nepal, only 34% of nursing students considered that e-learning was as efficient as traditional classroom learning, although 58.9% took a favorable attitude toward e-learning [19]. However, the results of other works conducted with health sciences students were like those herein reported, with most students feeling satisfied and with similar mean scores to our students [16,17]. In our study, moderate student satisfaction predominated (55.6%), with high student satisfaction obtained for slightly over one third of the sample (34.8%).

Several items with low ratings in our study, such as lack of Internet connection and teacher limitations with using digital platforms, coincide with the reasons for dissatisfaction cited in some studies, such as no ICT availability or connectivity problems [33] which, according to other authors, evidence socioeconomic inequalities and explain territorial differences [32,34].

Our results revealed moderate satisfaction for both online learning and b-learning, despite the overall evaluation being higher for b-learning. Similar results have been reported in another study within the European framework (Croatian Health Sciences students) with a satisfaction evaluation for exclusive e-learning of 3.7 (on a scale out of 5) [16]. Unequal results have been reported for variations in satisfaction according to gender: no significant gender differences were found in a Croatian study [16], while others concluded that the female sex is a predictor of satisfaction [17,18]. This falls in line with what our results found because the males in our study indicated feeling less satisfied with the online methodology than females (*p* = 0.003), but these differences were not confirmed for the mixed methodology (b-learning). Differences in satisfaction did not appear for students’ ages, but for courses. In our case, the difference in student satisfaction appeared only between courses 2 and 3 with the mixed methodology (*p* = 0.002). Other studies in health sciences (medicine and nursing) indicate differences in satisfaction between different courses [13], which coincides with our findings.

Factors related to feeling satisfied with online learning have been found, such as: correlations with learning commitment [23]; technology being available; good connections [13,14,33]. The students in the study from India were satisfied with the support and answers obtained from their faculties for setting up virtual classes, but were dissatisfied with other different aspects [13]. Likewise, students positively valued the adaptive efforts made in their centers after the pandemic was announced. N.M. Almusharraf reported that students were satisfied with university and faculty staff members. They agreed with not only the specific online platforms to be used, but also with the employed marking system, evaluation options, training workshops, online technical support and e-learning platforms [31].

Some of the most valued aspects in our study coincide with the proposals made for improvement cited in the literature on satisfaction with university studies; for example, good communication with faculties, their feedback and good communication with peers via the virtual learning environment [29].

Student preferences tend to include the combination of face-to-face learning (classic classroom learning) and e-learning [16,32], along with improvement proposals for a methodology in the future, and the combination of practical clinical learning and other methodologies. One feasible option could be face-to-face classes, practical sessions and online learning [13]. The need to develop plans to prepare emergency education that also include students’ well-being by diversifying methodologies and improving the use of digital technologies [15] has been pointed out [22]. Several works have proposed taking measures against the negative impacts of COVID-19 on health sciences students. They range from reducing stressful factors to supporting training by online learning and improving technologies to materializing them [35], improving students’ resilience and bearing in mind the diversity of students’ personal situations (economic, occupational or family). Our results could contribute to explore improvement proposals in the learning methodologies under study.

Although our study was initially population-based, the study population subset (effective sample) was formed according to students’ willingness to participate or not. This could imply a selection bias. Another limitation of this study could be the employed measuring instrument not being validated in the pandemic context that it was applied to. The herein utilized CUSAUF survey is validated for studies chosen with prior knowledge of the methodology (online, face-to-face, mixed) in the university setting. This context is exceptional because methodologies have been supervened and imposed by pandemic restrictions. Evidently, this aspect could affect perceived student satisfaction. Finally, another limitation could include the difficulty of extrapolating results to other university studies apart from the health sciences area.

A key contribution of our work would be that both methodology types would be satisfactory for our students to continue with university teaching in exceptional situations. Notwithstanding, the b-learning teaching system that included the face-to-face relationship was considered more satisfactory by students. This would show that students would feel satisfied with online teaching systems but, in turn, these students would demand face-to-face teaching and contact with the teaching staff.

Finally, future research could evaluate university student satisfaction with the evaluated teaching methodologies (e-learning and b-learning) in other degrees and in different situations, without imposed methodologies or supervening changes. It would also be interesting to test whether student and faculty computer competencies are factors that influence student satisfaction.

## 5. Conclusions

The nursing student satisfaction level with the learning received during two courses affected by methodological changes due to the COVID-19 pandemic was moderate at our university.

The satisfaction level was slightly higher during the course with the b-learning methodology compared to the semester with the exclusively online methodology set up when the pandemic began.

During the course with the b-learning methodology, the second year students were more satisfied with the teaching.

In our case, their satisfaction with the methodology, which included a certain degree of face-to-face attendance and teacher-discussant contact, was higher.

However, it is necessary to include more studies in different conditions, and without methodologies being imposed.

## Figures and Tables

**Table 1 healthcare-10-00597-t001:** Participants’ socio-demographic characteristics.

	Online Learning (March–June 2020) *n* = 139	B-Learning (January–February 2021) *n* = 101	*p* (t/X^2^)
Variables	Mean (SD)/*n* (%)	Mean (SD)/*n* (%)	
Age (years)	20.33 (3.1)	21.23 (4.80)	0.80
18–22	127 (91.4)	86 (85.1)	
23–27	7 (5.0)	10 (9.9)	
28–32	4 (2.9)	2 (2.0)	
+33	1 (0.7)	3 (3.0)	
Sex			0.79
Female	120 (86.3)	87 (86.5)	
Male	19 (13.7)	14 (13.5)	
University academic course			0.001 ^1^
First course	63 (45.3)	-	
Second course	40 (28.8)	60 (59.4)	
Third course	36 (25.9)	41 (40.6)	
Students’ computer resources			0.481
Own resources	136 (97.8)	100 (99.0)	
Resources provided by the university	3 (2.2)	1 (1.0)	
Resources provided by others	0 (0.0)	0 (0.0)	

Data are expressed as the mean with its standard deviation or their *n* and percentage. The association between learning methodology and sex, university academic course was analyzed with the X^2^-test. The association between learning methodology and age was analyzed with Student’s t. There are no data for the first academic course for the b-learning methodology. Statistical values: ^1^ (X^2^ Statistical = 62.884; df = 2) SD: Standard deviation; df: degrees of freedom.

**Table 2 healthcare-10-00597-t002:** Bivariate analysis. Relation between dimensions on the satisfaction scale and nursing students’ satisfaction with the learning methodology.

	Online Learning (March–June 2020) (*n* = 139)	B-Learning (January–February 2021) (*n* = 101)	*p* (t/X^2^)
Variables	Mean (SD)/*n* (%)	Mean (SD)/*n* (%)	
Nursing students’ overall satisfaction ^a^	2.73 (0.58)	2.94 (0.49)	0.004 ^1^
Nursing students’ satisfaction			0.07
High (>3 points)	48 (34.5)	32 (31.7)	
Moderate (2–3 points)	77 (55.4)	66 (65.3)	
Low (<2 points)	14 (10.1)	3 (3.0)	
Dimensions on the satisfaction scale ^a^			
Subjects’ structure	2.61 (0.70)	3.01 (0.58)	0.001 ^2^
Teacher-related aspects (teaching performance)	2.74 (0.59)	2.94 (0.54)	0.007 ^3^
Aspects related to the contents of the different subjects on the whole	2.60 (0.66)	2.92 (0.54)	0.001 ^4^
Communication-related aspects	2.91 (0.85)	3.05 (0.63)	0.147
Aspects related to the virtual learning environment	3.05 (0.64)	3.04 (0.56)	0.971

Data are expressed as the mean with its standard deviation or with their *n* and percentage. t: Student’s t statistical. The association between learning methodology and the nursing student satisfaction level was analyzed with the X^2^-test. The association between learning methodology and the dimensions on the satisfaction scale and overall nursing student satisfaction was analyzed with Student’s t. There are no data for the first academic course for the b-learning methodology. ^a^: The student satisfaction level score on a Likert-type scale from 1 to 4. Statistical values:^1^ (Student’s t = −2.915; df: 238; 95% CI −0.35–−0.07) ^2^ (Student’s t = −4.869; df: 231.75; 95% CI −0.57–0.24) ^3^ (Student’s t = −2.705; df: 231.17; 95% CI −0.36–−0.06) ^4^ (Student’s t = −3.960; df 221.32; 95% CI −0.47–−0.15) df: degrees of freedom; SD: standard deviation; df: degrees of freedom.

**Table 3 healthcare-10-00597-t003:** Bivariate analysis. Covariates and nursing student satisfaction (online learning and b-learning.

	Nursing Student Satisfaction/ Online Learning (June 2020) (*n* = 139)				Nursing Student Satisfaction/ b- Learning (January–February 2021) (*n* = 101)			
	High	Moderate	Low	*p* (X^2^/F)	High	Moderate	Low	*p* (X^2^/F)
Covariates	Mean (SD)/*n* (%)	Mean (SD)/*n* (%)	Mean (SD)/*n* (%)		Mean (SD)/*n* (%)	Mean (SD)/*n* (%)	Mean (SD)/*n* (%)	
Age (years)	20.35 (2.99)	20.53 (3.40)	19.14 (1.23)	0.309	21.72 (6.11)	21.02 (4.17)	20.67 (1.15)	0.780
Sex					0.003 ^1^			0.659
Female	44 (36.7)	68 (56.7)	8 (6.7)		28 (32.6)	55 (64.0)	3 (3.5)	
Male	4 (21.1)	9 (47.4)	6 (31.6)		4 (26.7)	11 (73.3)	0 (0.0)	
University academic course				0.297				0.002 ^2^
First course	26 (41.3)	29 (46.0)	8 (12.7)		-	-	-	
Second course	12 (30.0)	26 (65.0)	2 (5.0)		26 (43.3)	34 (56.7)	0 (0.0)	
Third course	10 (27.8)	22 (61.1)	4 (11.1)		6 (14.6)	32 (78.0)	3 (7.3)	
Students’ computer resources				0.389				0.337
Own resources	47 (34.8)	75 (55.6)	13 (9.6)		31 (31.0)	66 (66.0)	3 (3.0)	
Resources provided by the university	1 (33.3)	1 (33.3)	1 (33.3)		1 (100.0)	0 (0.0)	0 (0.0)	
Resources provided by others	0 (0.0)	0 (0.0)	0 (0.0)		0 (0.0)	0 (0.0)	0 (0.0)	

Data are expressed as the mean with its standard deviation or with their *n* and percentage. The association between nursing student satisfaction and sex, university academic course and students’ computer resources were analyzed with the X^2^-test. The association between nursing student satisfaction and age was analyzed with ANOVA. The value of the used statistic is indicated below the level of significance. For bimodal learning, no data were found for the first academic course. Statistical values: ^1^ (X^2^ Statistical = 11.521; df = 2) ^2^ (X^2^ Statistical = 12.426; df = 2) SD: standard deviation; df: degrees of freedom.

**Table 4 healthcare-10-00597-t004:** Nursing student satisfaction with online and bimodal learning per course measured with the adapted CUSAUF questionnaire. Bivariate analyses.

	University Academic Course				
	First	Second	Third	Total	*p*
Variables	Mean (SD); [IC95%]	Mean (SD); [IC95%]	Mean (SD); [IC95%]	Mean (SD); [IC95%]	
Nursing students’ overall Satisfaction ^a^					
Online learning (first wave)	2.81 (0.60) [IC 95%: 2.66–2.97]	2.69 (0.52) [IC 95%: 2.53–2.86]	2.61 (0.61) [IC 95%: 2.40–2.82]	2.73 (0.58) [IC 95%: 2.63–2.83]	0.224
Bimodal learning (second wave)	-	3.07 (0.49) [IC 95%: 2.95–3.20]	2.74 (0.44) [IC 95%: 2.60–2.88]	2.94 (0.49) [IC 95%: 2.84–3.03]	0.001 ^1^
Satisfaction dimensions and learning type ^a^					
Subjects’ structure					
Online learning (first wave)	2.67 (0.78) [IC 95%: 2.47–2.87]	2.55 (0.60) [IC 95%: 2.35–2.75]	2.55 (0.65) [IC 95%: 2.33–2.78]	2.61 (0.70) [IC 95%: 2.49–2.73]	0.622
Bimodal learning (second wave)	-	3.16 (0.58) [IC 95%: 3.01–3.31]	2.73 (0.46) [IC 95%: 2.64–2.95]	3.01 (0.58) [IC 95%: 2.90–3.13]	0.001 ^2^
Teacher-related aspects (teaching performance) ^a^					
Online learning (first wave)	2.80 (0.60) [95% CI: 2.64–2.95]	2.73 (0.49) [95% CI: 2.56–2.90]	2.64 (0.66) [95% CI: 2.41–2.87]	2.74 (0.59) [95% CI: 2.64–2.84]	0.445
Bimodal learning (second wave)	-	3.10 (0.54) [95% CI: 2.96–3.25]	2.73 (0.46) [95% CI: 2.58–2.87]	2.94 (0.54) [95% CI: 2.83–3.05]	0.001 ^3^
Aspects related to the contents of the different subjects on the whole ^a^					
Online learning (first wave)	2.69 (0.64) [95% CI: 2.53–2.86]	2.54 (0.64) [95% CI: 2.34–2.75]	2.51 (0.72) [95% CI: 2.25–2.77]	2.60 (0.66) [95% CI: 2.49–2.72]	0.350
Bimodal learning (second wave)	-	3.11 (0.49) [95% CI: 2.98–3.24]	2.66 (0.48) [95% CI: 2.51–2.81]	2.92 (0.53) [95% CI: 2.81–3.03]	0.001 ^4^
Communication-related aspects ^a^					
Online learning (first wave)	2.93 (0.88) [95% CI: 2.71–3.16]	2.90 (0.73) [95% CI: 2.66–3.13]	2.87 (0.94) [95% CI: 2.56–3.19]	2.91 (0.85) [95% CI: 2.76–3.05]	0.940
Bimodal learning (second wave)	-	3.14 (0.64) [95% CI: 2.98–3.24]	2.92 (0.60) [95% CI: 2.73–3.12]	3.05 (0.63) [95% CI: 2.93–3.18]	0.093
Virtual learning environment-related aspects ^a^					
Online learning (first wave)	3.20 (0.47) [95% CI: 3.08–3.32]	3.00 (0.72) [95% CI: 2.77–3.24]	2.82 (0.75) [95% CI: 2.55–3.08]	3.05 (0.64) [95% CI: 2.94–3.16]	0.017 ^5^
Bimodal learning (second wave)	-	3.14 (0.56) [95% CI: 2.99–3.29]	2.91 (0.55) [95% CI: 2.73–3.08]	3.04 (0.56) [95% CI: 2.93–3.16]	0.046 ^6^

Values are expressed as the mean with standard deviation in brackets and the 95% confidence interval. First-year students did not follow the b-learning methodology system. F: Fisher’s F statistic. ^a^: Student satisfaction level for scores on a Likert-type scale from 1 to 4. Statistical values: ^1^ (Fisher’s F statistic = 12.255; df = 1) ^2^ (Fisher’s F statistic = 10.788; df = 1) ^3^ (Fisher’s F statistic = 12.861; df = 1) ^4^ (Fisher’s F statistic = 19.762; df = 1) ^5^ (Fisher’s F statistic= 4.193; df = 2) ^6^ (Fisher’s F statistic= 4.084; df = 1) SD: standard deviation; df: degrees of freedom.

## Data Availability

Not applicable.

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
