# Peer review of "Nursing Student Satisfaction with the Teaching Methodology Followed during the COVID-19 Pandemic"

_healthcare, 2022, doi:10.3390/healthcare10040597_

Round 1
Reviewer 1 Report
The work discusses the nursing students perception of satisfaction in learning due to the pandemic. The scale and approach looks reasonable, but I have the following questions:
1) For the Likert scale, it is not sure on what scale it is on, and why that scale is chosen?
2) From the discussion, it appears that much of the conclusions derived from the survey have been explored in earlier works. In this case, what would be the key contributions of this work, other than validating the earlier reports?
Reviewer 2 Report
Dear authors, here you can find some suggestions for your manuscript.

Round 2
Reviewer 2 Report
Dear authors, thank you for your new version, it has improved very much, congratulations on your hard work. I don't have any more suggestions for it. Best regards